# Better Surface Integrity and Tribological Properties of Steel Sintered by Powder Metallurgy

**DOI:** 10.3390/ma13143172

**Published:** 2020-07-16

**Authors:** Tae-Hwan Lim, Chang-Soon Lee, In-Sik Cho, Auezhan Amanov

**Affiliations:** 1Department of Advanced Materials Science, Sun Moon University, Asan 31460, Korea; ltw@sunmoon.ac.kr (T.-H.L.); lcs8769@sunmoon.ac.kr (C.-S.L.); mbrosia1018@naver.com (I.-S.C.); 2Department of Mechanical Engineering, Sun Moon University, Asan 31460, Korea

**Keywords:** steel, surface integrity, porosity, tribology, powder metallurgy, ultrasonic nanocrystal surface modification

## Abstract

The current research reports the improvement in surface integrity and tribological characteristics of steel prepared using a powder metallurgy (PM) by ultrasonic nanocrystal surface modification (UNSM) at 25 and 300 °C. The surface integrity and tribological properties of three samples, namely, as-PM, UNSM-25 and UNSM-300 were investigated. The average surface roughness *(R_a_)* of the as-PM, UNSM-25 and UNSM-300 samples was measured using a non-contact 3D scanner, where it was found to be 3.21, 1.14 and 0.74 µm, respectively. The top surface hardness was also measured in order to investigate the influence of UNSM treatment temperature on the hardness. The results revealed that the as-PM sample with a hardness of 109 HV was increased up to 165 and 237 HV, corresponding to a 32.1% and 57.2% after both the UNSM treatment at 25 and 300 °C, respectively. XRD analysis was also performed to confirm if any changes in chemistry and crystal size were took place after the UNSM treatment at 25 and 300 °C. In addition, dry tribological properties of the samples were investigated. The friction coefficient of the as-PM sample was 0.284, which was reduced up to 0.225 and 0.068 after UNSM treatment at 25 and 300 °C, respectively. The wear resistance was also enhanced by 33.2 and 52.9% after UNSM treatment at both 25 and 300 °C. Improvements in surface roughness, hardness and tribological properties was attributed to the elimination of big and deep porosities after UNSM treatment. Wear track of the samples and wear scar of the counter surface balls were investigated by SEM to reach a comprehensive discussion on wear mechanisms. Overall, it was confirmed that UNSM treatment at 25 and 300 °C had a beneficial effect on the surface integrity and tribological characteristics of sintered steel by the PM that is used in a shock absorber for a car engine.

## 1. Introduction

The sintered components by the powder metallurgy (PM) process are manufactured through the process of molding and sintering by mixing raw powders, so they can form complex shapes and have strengths in terms of productivity, and have grown significantly in a relatively short period of time. In particular, it has been actively applied over many decades to automobile components as an optimal material to realize cost and weight reductions [1]. Physical properties of elastic modulus and Poisson’s ratio were measured using a shock absorber sintered component for an automobile engine. In the case of the piston of the shock absorber, it is important to improve the machinability efficiency because there is a process of making a product by machining a hole due to the characteristics of the components made of steel. Accordingly, an MnS element was added to steel for improving machinability. However, the presence of MnS improves machinability of the sintered components, but the mechanical properties and tribological characteristics are still poor. The relationship between machinability, and mechanical properties and tribological characteristics can be explained that when the material has a good machinability that may lead to the lower cutting forces and better surface quality. However, the percentage of added MnS inclusions is too low, that is why they are not critical for the component performance. In this regard, it is important to improve the strength and tribological characteristics of steel containing MnS inclusions. Recently, a demand on new materials with high strength and better tribological characteristics has increased [2]. However, there is no way to satisfy such a demand with porosity materials.

Accordingly, in the case of the PM process, micropores are changed by microinclusions. Although it is inevitable to increase the porosity for the functionality of the ferrous material, it is inevitable that the strength decreases as the pores increase, and the brittleness increases, so that both the porosity and strength can be matched due to damage caused by the increase in pores [3]. Therefore, in this study, the surface integrity, hardness and tribological characteristics of sintered steel for automobile engines were investigated. The machinability of sintered steels is influenced by several factors, including chemical composition, microstructure, inclusions and thermomechanical properties. Accordingly, the methods of strengthening are also divided into various ways, and experimental research on strengthening is concentrated.

With the aim of strengthening the PM materials a surface severe surface plastic deformation (S^2^PD) is an excellent method as it is capable of increasing the hardness by closing up porosities. In this study, an ultrasonic nanocrystal surface modification (UNSM), which is an S^2^PD technology, is introduced. According to the previous study [4] it is strongly believed that this technology may lead to better surface integrity, hardness and tribological properties of sintered steel by PM. However, an attention needs to be paid to the UNSM treatment parameters are selected based on the initial properties and surface condition of the material because of the appropriate UNSM parameters. Amanov et al. applied a UNSM technology to Cu alloy prepared by PM for automobile applications [5]. Interestingly, the UNSM technology increased the strength owing to the presence of the nanocrystal surface layer, whose grains were within the nanoscale, and also disappearance of grain boundaries. Again the same authors evaluated the effect of UNSM technology on tribological and scratch characteristics of the Cu alloy prepared by PM [6]. Dry tribological behavior along with scratch resistance was also enhanced in comparison with the untreated Cu alloy. However, the UNSM has never been applied to the steel containing MnS inclusions at room and high temperatures (RT and HTs), while the UNSM is able to increase the strength of various conventional steels at both RT and HTs, but it sometimes leads to softening due to grain growth during heating.

Hence, it is a challenge to apply the UNSM technology to the steel by PM containing MnS inclusions. The main objective of this investigation is to check if the UNSM could enhance the strength of steel containing MnS inclusions. If yes then subsequently evaluate its effects on tribological characteristics. The findings from this study will be beneficial not only to the shock absorber for car engine but also to widening the applications of steel containing MnS inclusions in different potential industries. Enhancement mechanisms of tribological properties of steel by PM before and after UNSM treatment were discussed comprehensively in order to understand the role of UNSM technology.

## 2. Materials and Experimental Methods

### 2.1. Material Preparation by PM

A Fe-based high C sintered steel used in this study, a ring-shaped molded samples with a diameter of 30 mm, an inner diameter of 15 mm and a thickness of 10 mm were prepared. This type of samples with a density of 6.62 g/cm^3^ is usually used for mass production using an 800-ton class automatic press. The sintering was performed at a temperature of 1120 °C for 20 min in an ENDO-deformed gas atmosphere using a mesh belt sintering furnace. The cooling rate after sintering was 6.0 °C/min. The chemical composition of the sintered Fe-based high C sintered steel is listed in Table 1. A picture of the sintered sample can be seen in Figure 1.

### 2.2. UNSM Treatment

UNSM treatment is one of the new mechanical surface modification cold-forging processes. The main principal of UNSM treatment is to introduce an S^2^PD, as illustrated in Figure 2a, that leads to grain size refinement and high compressive residual stress at the surface and subsurface layers, whereas increasing mechanical properties of metallic materials. UNSM treatment parameters are provided in Table 2. WC Dymet grade DF3 (ISO Code K01) was used in the UNSM treatment. The wear WC ball was not taken into account due to its high hardness of 2000 HV compared to the sintered steel. The samples were polished first and then subsequently treated by UNSM treatment at 25 and 300 °C. The temperature of the sample was increased by using a halogen lamp as illustrated in Figure 2b. It is important to mention that the effect of UNSM technology on the dimensional control of the components was very negligible. Details of UNSM treatment can be found in our previous studies [4,5,6].

### 2.3. Tribological Test

Tribological characteristics of the as-PM, UNSM-25 and UNSM-300 samples were evaluated using a tribometer (CSM Instruments, Peseux, Switzerland) against SAE 52100 bearing steel with a diameter of 7.14 mm under dry conditions. Tribological test conditions are listed in Table 3. The tribological experiments were replicated three times for each samples due to data scattering.

### 2.4. Characterizations

The surface and cross-sections and wear track of the sample, and also wear scar of the counter surface ball were observed by a scanning electron microscope (SEM) along with an energy-dispersive X-ray spectroscope (EDX) was used (JSM-6610LA, JEOL, Tokyo, Japan). The surface roughness and hardness were measured 3–4 times by using a profilometer (SJ-210, Mitutoyo, Tokyo, Japan) and a micro-Vickers tester at 300 gf and 12 s (MVK E3, Mitutoyo, Tokyo, Japan), respectively. Microstructural changes were analyzed using an X-ray diffraction (XRD: D8 ADVANCE, Bruker, Karlsruhe, Germany) with a chromium (CrKα) radiation (2.2897 Å) at a wavelength of 0.210. Compressive residual stress was measured by an X-ray method (HyPix-3000, Rigaku, Tokyo, Japan). Grain size refinement was analyzed by a transmission electron microscope (TEM: Titan Themis Z, FEI, Hillsboro, OR, USA) operated at 300 kV and the size of at least 20 nano-grains were measured by using an ImageJ version 1.51 (NIH, Bethesda, MD, USA) software. The samples were washed in a solution of 90% ethanol and 10% nitric acid at a voltage of 5 V for 3 min to keep surfaces visibly clean and free from contaminants. Wear tracks generated on the surface of the samples after tribological testing were analyzed with the help of Vision64 software provided by 3D stylus profilometry (Dektak XT, Bruker, Karlsruhe, Germany).

## 3. Results and Discussions

### 3.1. Morphology

A comparison of SEM images for the as-PM, UNSM-25 and UNSM-300 samples are shown in Figure 3. The as-PM sample had a number of pores on the surface (see Figure 3a) that led to high surface roughness and low hardness values. Those pores did not disappear from the surface after polishing with a SiC sandpaper. While the existing pores on the surface were closed down in part by UNSM at RT and HTs as shown in Figure 3b,c, but it was not able to close down large-sized pores due to the imperfection of the impact of UNSM treatment. By increasing the UNSM treatment temperature it was discovered that the size of unclosed porosities after UNSM at 25 °C were further diminished after UNSM at 300 °C. The UNSM-induced parallel lines were observed on the surface as shown in Figure 4b,c, where the distance was about 70 µm, which is the result of the feed-rate listed in Table 2. It can be assessed based on the previous study that UNSM technology can reduce the porosities of powder-based metal additive manufacturing (AM) [7]. As a result, the UNSM treatment at both 25 and 300 °C improved the surface finish with less porosities compared to that of the as-PM sample, but the surface finish was further advanced by increasing the temperature of the UNSM treatment.

### 3.2. Roughness

The results of arithmetic average roughness (*R_a_*) surface roughness measurements of the as-PM, UNSM-25 and UNSM-300 samples are shown in Figure 5. It can be seen from Figure 4a that the as-PM sample had a high *R_a_* value of 3.21 μm, where the samples after UNSM treatment at both 25 and 300 °C *R_a_* was significantly reduced up to 0.114 and 0.74 μm, which was attributed to direct bombardment of the UNSM tip on the sample surface, respectively. In the high temperature state, it is considered that during the heat treatment, the surface layer was deformed significantly in comparison with the UNSM treatment at the 25 °C state, where the static load was applied through the moving path interval value set by a computer numerical control (CNC) machine. In the case of the as-PM sample, pores generated during the sintering were observed as shown in Figure 5, but it was confirmed that the deep pores formed on the surface were disappeared after UNSM treatment at both 25 and 300 °C as shown in Figure 5. It is generally known that the higher surface roughness can cause dehancement in the tribological performance and provide crack initiation, which eventually leads to fatigue. For example, mechanical surface milling that had the lowest surface roughness values, also showed the highest improvement in the tribological performance [8]. Considering the roughness effect for improving tribological performance, it is advantageous to include UNSM treatment. Consequently, reduction in high peaks and valleys after UNSM treatment at both 25 and 300 °C is responsible for the reduction of surface roughness as can be seen in Figure 5. In addition, the disappearance of pores depending on the size and deepness played a significant role in controlling the surface roughness of steel produced by PM. 

### 3.3. Hardness, Residual Stress and Grain Size

Figure 6 shows the results of hardness measurements with respect to the depth from the surface of the as-PM, UNSM-25 and UNSM-300 samples. The average value was obtained after four measurements. The surface hardness of the as-PM thereof was increased from 109 to 165 HV and 237 HV for the UNSM-25 and UNSM-300 samples, respectively. In general, depth profiles of modification layers may reach up to about 200 µm depending on UNSM treatment parameters. The indentation depth at a 300 gf indentation force was about 35–38 µm, which was remained within the depth of the nanocrystal structure. In the case of UNSM technology, since the surface layer is modified with a nanocrystal structure, it is thought that the hardness increased due to the refinement of the surface layer grains. One of the important properties of as-PM materials is porosity that can control the surface hardness [9]. The effective hardened depth of the UNSM-300 sample was supposed to be the deepest and highest hardness result. It is shown that UNSM can improve the surface hardness by increasing dislocation density and repeated bombardment may have a beneficial effect on surface hardening. The increase in hardness is mainly attributed to the refinement of coarse grains into nano-sized grains according to the Hall–Petch expression [10]. In addition, work hardening and reduction in the number of porosities are responsible for it.

The residual stress of the as-PM sample was found to be approximately −21 MPa, while the UNSM treatment at 25 and 300 °C introduced compressive residual stress of −418 and −637 MPa, respectively. It was confirmed that the level of introduced compressive residual stress increased by increasing the temperature of UNSM treatment. Compressive residual stress is dominant regarding crack formation and propagation. In order to define the effect of the UNSM treatment temperature on grain size TEM was employed. Top surface bright-field TEM images of the UNSM-25 and UNSM-300 samples are shown in Figure 7. It was observed that the UNSM technology produced a nanocrystal structure, where the grain size refined with increasing the temperature of UNSM technology. As a result, the average nano-grain size of the UNSM-25 and UNSM-300 samples was found to be approximately 167 and 114 nm, which was in agreement with the Hall–Petch relationship expressing the smaller the crystal size, the higher the hardness.

### 3.4. XRD Results

XRD patterns correspond to martensite derived from the as-PM, UNSM-25 and UNSM-300 samples are shown in Figure 8. No shift in XRD patterns was found, which implies that no phase transformation occurred after UNSM treatment at both RT and HT. Fortunately, a slight reduction in intensity was observed as shown in Figure 8, which implies the refinement of coarse grains into nano-sized grains after UNSM treatment at RT and HT. Full width at half maximum (FHWM) values of the martensitic (200) peaks was calculated as listed in Table 4. It was confirmed that those FHWM values were got broadened after UNSM treatment at both RT and HT compared to that of the as-PM sample, which is the response of the nanocrystal surface layer [5]. In general, the depth of the nanocrystal surface layer generated by UNSM technology can be reached up to 200 µm [4].

### 3.5. Tribological Characteristics 

A change in the friction coefficient with respect to the sliding time of the as-PM, UNSM-25 and UNSM-300 samples is shown in Figure 9. In the case of the as-PM sample, the friction coefficient increased up to 0.315 and a bit reduced up to 0.284 with high fluctuation. In the case of UNSM-25, the friction coefficient increased gradually from 0.067 to 0.224 within a sliding time of 3000 sec, which implies a running-in period and afterwards by the end of the test it was stabilized with a value of 0.225 with high fluctuation. In the case of UNSM-300, the friction coefficient was very stable with a value of 0.068 with a low fluctuation throughout the sliding time. Accordingly, the UNSM at 25 °C reduced the friction coefficient by approximately more than two times compared to that of the as-PM sample. Satisfyingly, the UNSM at 300 °C reduced the friction coefficient by approximately 80% and another more than two times compared to that of the as-PM and UNSM-25 samples, respectively. An abrupt increase in friction coefficient of the as-PM sample was due to the surface integrity after PM technology. Moreover, deliberate running-in and steady-state periods in the friction of the UNSM-25 sample can be attributed to the wear of asperities generated during several cycles. The effect of the UNSM treatment temperature was obviously confirmed where HT led to the better frictional behavior because of the low roughness and high hardness. It has been reported that the UNSM at RT improved the frictional behavior of PM materials due to aforementioned reasons [6].

Aside from frictional behavior, the wear resistance of the as-PM, UNSM-25 and UNSM-300 samples was evaluated as shown in Figure 10 based on the wear track dimensions obtained using a three-dimensional surface images (see Figure 11). It needs to be mentioned here that the depth of the wear tracks generated on the surface of both the UNSM-25 and UNSM-300 samples was within the depth of the nanocrystal layer [11]. In Figure 11, it can be observed that some porosities still existed within the wear track on the surface of the as-PM sample, while small-sized porosities existed within non-semispherical wear track on the surface of the UNSM-25 and UNSM-300 samples after a sliding time of 5000 s, respectively. Following the friction coefficient value, the wear resistance exhibited a similar trend, where it was enhanced by approximately 33.2 and 52.9% for the UNSM-25 and UNSM-300 samples compared to that of the as-PM one, respectively. Again, the UNSM-300 sample demonstrated better wear resistance compared to that of the UNSM-25 one. Hence, it leads to the conclusion that the effect of UNSM treatment on tribological behavior was obvious, where the tribological behavior of the UNSM-300 sample was found to be much better than that of the UNSM-25 one. Better tribological behavior of both the UNSM-25 and UNSM-300 samples is mainly attributed to the improvement in surface integrity and hardness [5], where the initial contact interface controls the tribological behavior. In addition, the application of UNSM at RT to PM materials was employed where the wear resistance got better than that of the as-PM materials [4].

### 3.6. Wear Track Analysis

SEM images of the wear tracks and wear scars formed on the surfaces of the as-PM, UNSM-25 and UNSM-300 samples together with their counterparts are shown in Figure 12. Undoubtedly, the dimensions of wear tracks and wear scars formed on the surfaces of the as-PM were much bigger compared to that of the UNSM-25 and UNSM-300 samples as shown in Figure 12a–f. To be specific, the width of the wear track and diameter of the wear scar formed on the surfaces of the as-PM were approximately 0.768 and 1.429 mm, respectively. Comparably, the width of the wear track and diameter of the wear scar formed on the surfaces of the UNSM-25 and UNSM-300 samples were approximately 0.512 and 0.667 mm, and 0.753 and 0.659 mm, respectively. The smaller size of the wear tracks and wear scars formed on the surfaces of the UNSM-25 and UNSM-300 samples is attributed to the partially elimination of porosities, increase in hardness and also a reduction in surface roughness [5]. A reduction in the roughness of the UNSM-25 and UNSM-300 samples by UNSM treatment may reduce the true contact interface between the samples and counterface ball in relative motion due to the waviness created due to the UNSM treatment scanning trajectory as shown in Figure 4. A huge difference in roughness before and after UNSM treatment is due to the presence and absence of porosities as shown in Figure 5.

Wear mechanisms of two different steels under dry sliding conditions were investigated using SEM images. It was revealed that as a dominant wear mechanism for all the samples was an abrasive mode and also an oxidative wear mode took place as shown in Figure 12a–f. However, the oxidation level was different for each sample. For example, the oxidation within the wear track generated on the surface of the as-PM sample occurred rarely, but basically it took place impressively at the edge of the wear track generated on the surface of the UNSM-25 sample (see Figure 12a). It means that an oxide tribo-layer was formed within the wear track, but it swept out towards the edge of the of the wear track. The oxidation occurred within the wear track generated on the surface of the UNSM-25 and UNSM-300 samples (see Figure 12c,e), but the formed oxide tribo-layer was destroyed due to the presence of waviness leading to the high pressure at the contact interface. Basically, the formed oxide tribo-layer tended to control the tribological behavior of the materials that come into contact, as it is a preventative of similar materials coming into contact [12]. The different level and distribution of oxidation within the wear track are related to the roughness and true contact area, where the oxidation layer cannot bear a high stress due to its thin thickness [13,14]. Moreover, it is evident from Figure 12b,d,f that the considerable oxide-based tribo-layer was formed on the surface of the counterface ball that came into contact with the UNSM-25 and UNSM-300 samples, while no oxide tribo-layer was formed on the surface of the counterface ball that came into contact with the as-PM sample. Figure 13a–c shows the high-magnification SEM images of wear track formed on the surface of the as-PM, UNSM-25 and UNSM-300 samples. It is clear from Figure 13a that porosity with a diameter of approximately 30 µm was within the wear track after sliding of 5000 s. In turn, some wear debris and nearly non-worn out areas are seen in Figure 13b,c, which is the result of corrugated surface after UNSM treatment.

## 4. Conclusions

This study dealt with the effect of UNSM treatment at 25 and 300 °C on surface integrity and tribological characteristics of steel prepared by PM. At the end of this manuscript the following conclusions may be drawn as a result of the research based on the experimental tests and measurements: Average surface roughness of the as-PM, UNSM-25 and UNSM-300 samples was 3.21, 1.14 and 0.74 µm, respectively.

Hardness of the as-PM sample with a hardness of 109 HV was increased after both the UNSM at 25 °C and UNSM treatment at 300 °C up to 165 and 237 HV, corresponding to a 32.1% and 57.2%, respectively.The level of introduced compressive residual stress increased by increasing the temperature of UNSM treatment from −418 to −637 MPa.The intensity derived from XRD peak of the as-PM sample was reduced after both the UNSM at 25 °C and UNSM treatment at 300 °C, which implies the refinement of coarse grains into nano-sized grains occurred.The size of nano-grains of the UNSM-25 and UNSM-300 samples was approximately 167 and 114 nm, respectively.The friction coefficient of the as-PM sample was 0.284, which was reduced after UNSM treatment at 25 and 300 °C up to 0.225 and 0.068, respectively.The wear resistance was also enhanced by 33.2 and 52.9% after UNSM treatment at both 25 and 300 °C. Improvements in the surface roughness, hardness and tribological properties was attributed to the elimination of big and deep porosities after UNSM treatment at 25 and 300 °C.Overall, it was confirmed that UNSM treatment at 25 and 300 °C had a beneficial effect on the surface integrity and tribological characteristics of sintered steel by PM that is used in a shock absorber for a car engine.

## Figures and Tables

**Figure 1 materials-13-03172-f001:**
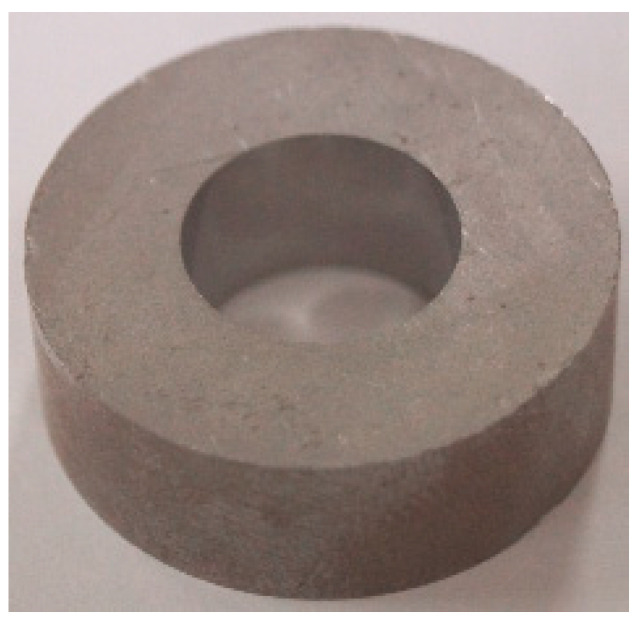
A picture of the sintered sample.

**Figure 2 materials-13-03172-f002:**
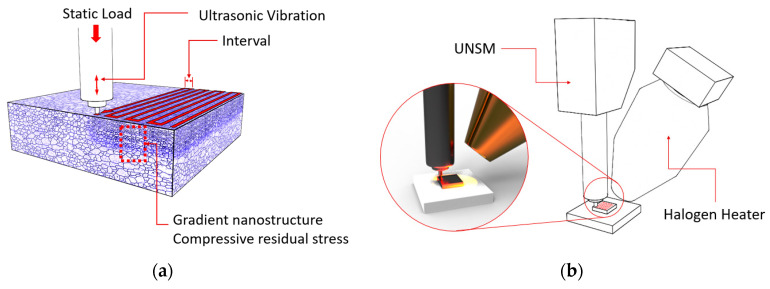
An illustration ultrasonic nanocrystal surface modification (UNSM) treatment showing the modified surface and subsurface of the sample (**a**) and the application of a heat by halogen lamp (**b**).

**Figure 3 materials-13-03172-f003:**
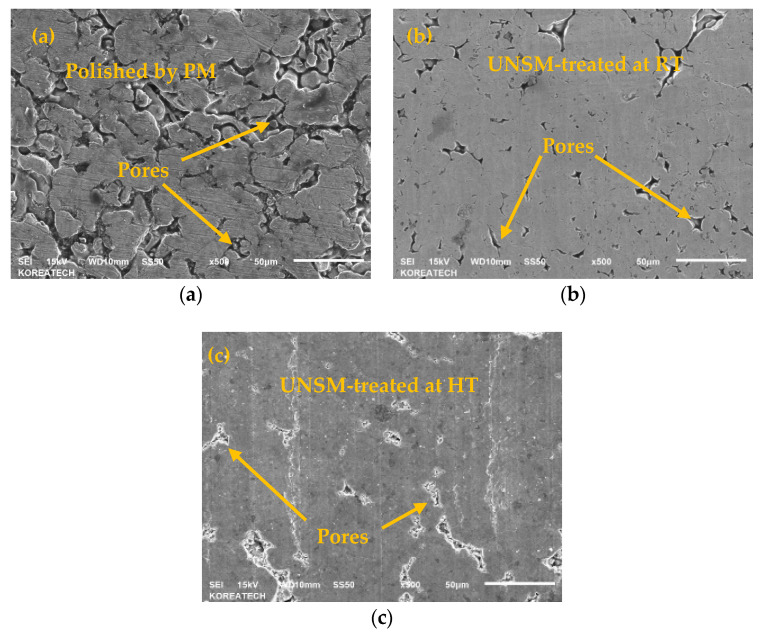
SEM images of the as-PM (**a**), UNSM-25 (**b**) and UNSM-300 (**c**) samples showing surface morphology.

**Figure 4 materials-13-03172-f004:**
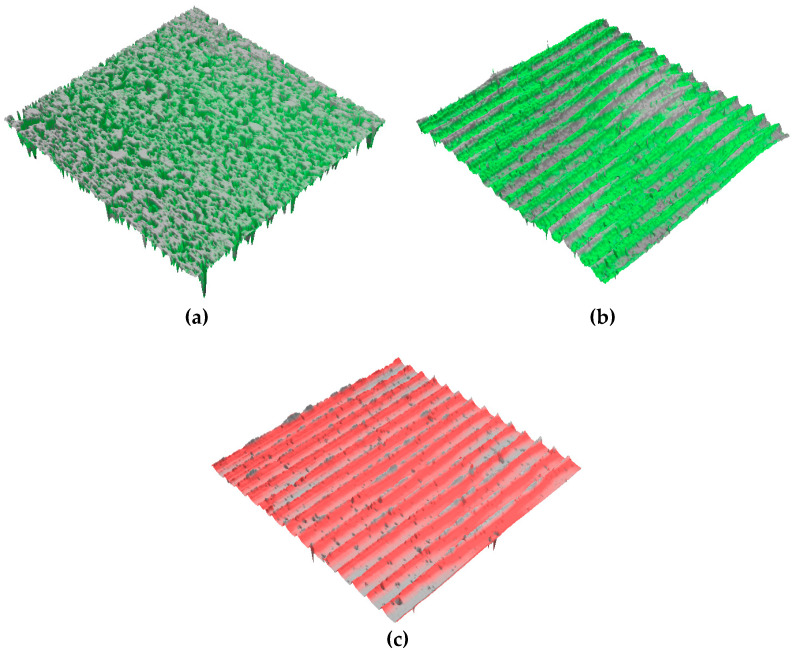
3D images of the as-PM (**a**), UNSM-25 (**b**) and UNSM-300 (**c**) samples showing surface morphology (scan area: 2.0 mm × 2.0 mm).

**Figure 5 materials-13-03172-f005:**
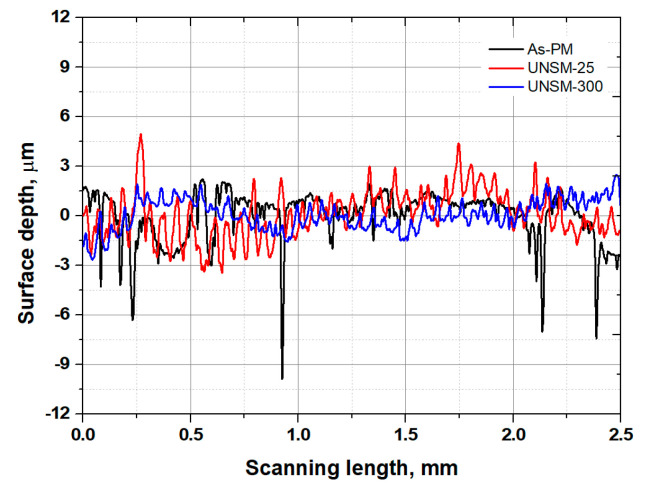
Comparison in surface roughness results of the as-PM, UNSM-25 and UNSM-300 samples.

**Figure 6 materials-13-03172-f006:**
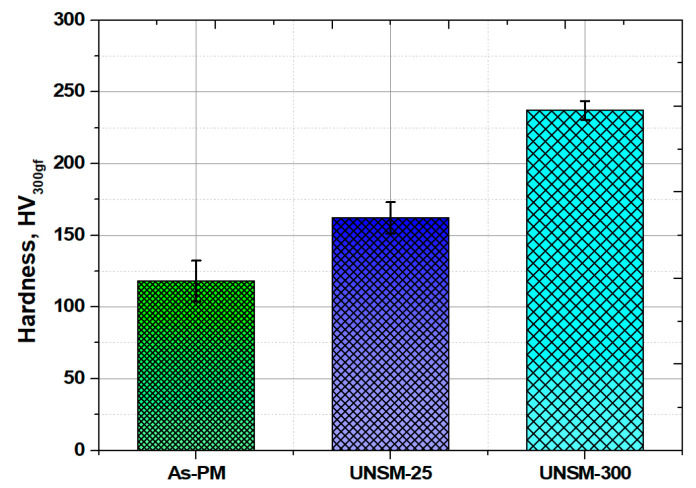
Comparison in surface hardness of the as-PM, UNSM-25 and UNSM-300 samples.

**Figure 7 materials-13-03172-f007:**
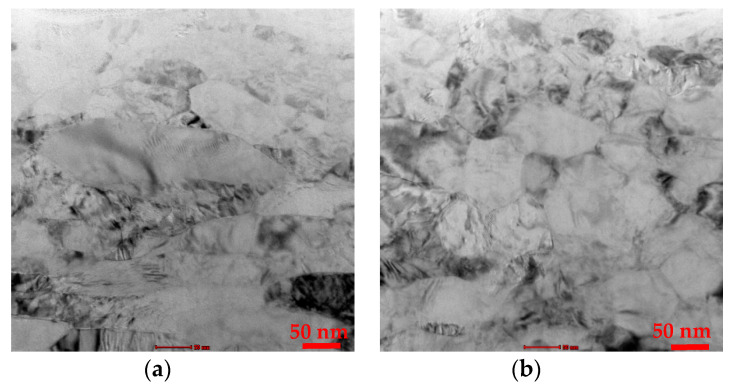
Bright-field top surface TEM images of the UNSM-25 (**a**) and UNSM-300 (**b**) samples.

**Figure 8 materials-13-03172-f008:**
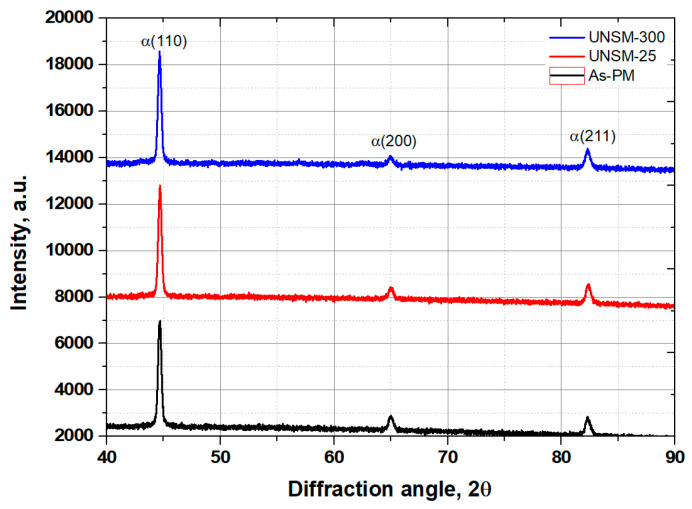
XRD patterns derived from the as-PM, UNSM-25 and UNSM-300 samples.

**Figure 9 materials-13-03172-f009:**
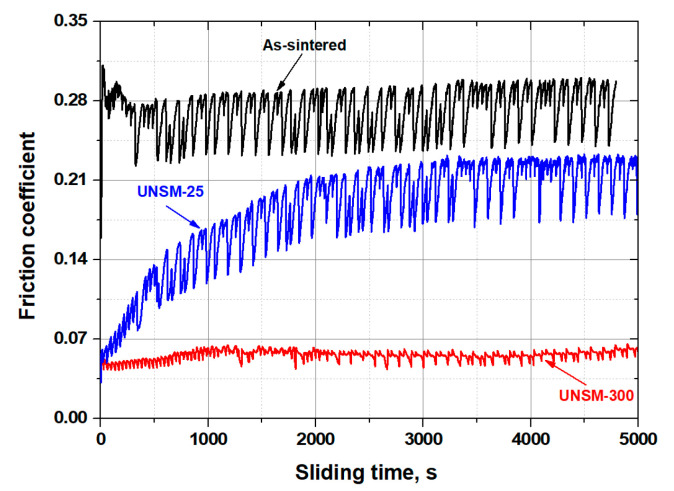
Friction coefficient results of the as-PM, UNSM-25 and UNSM-300 samples.

**Figure 10 materials-13-03172-f010:**
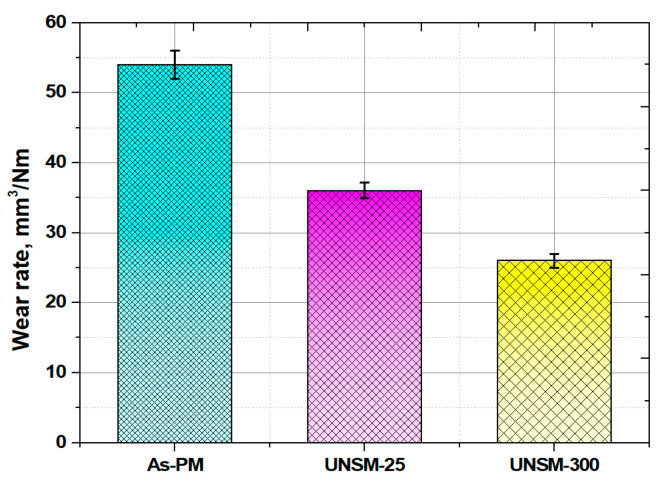
Wear resistance of the as-PM, UNSM-25 and UNSM-300 samples.

**Figure 11 materials-13-03172-f011:**
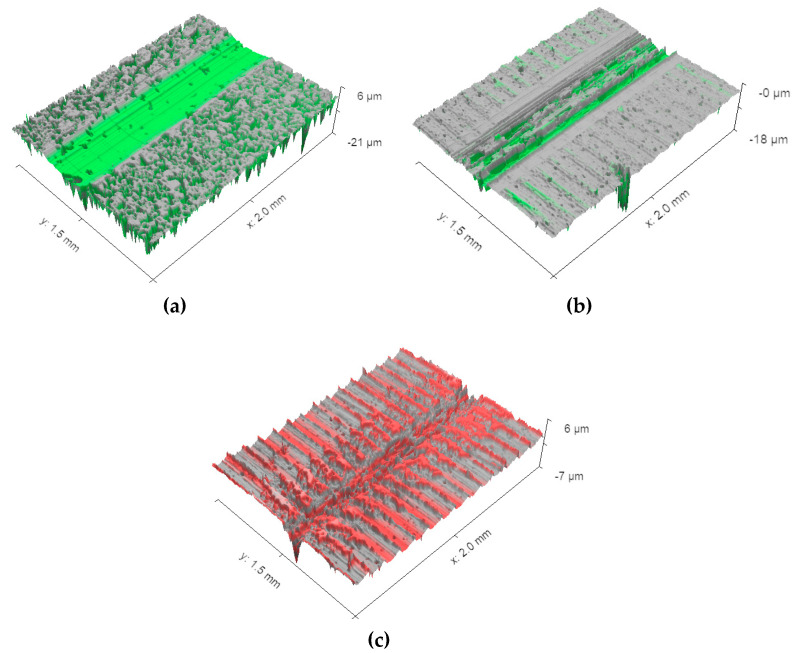
3D wear track images of the as-PM (**a**), UNSM-25 (**b**) and UNSM-300 (**c**) samples.

**Figure 12 materials-13-03172-f012:**
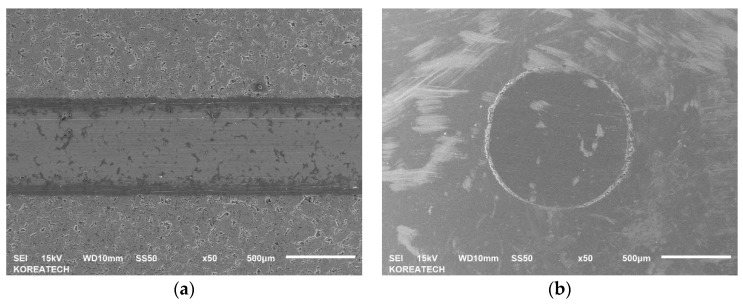
SEM images and mappings of the wear track formed on the surface of the as-PM (**a**,**b**), UNSM-25 (**c**,**d**) and UNSM-300 (**e**,**f**) samples.

**Figure 13 materials-13-03172-f013:**
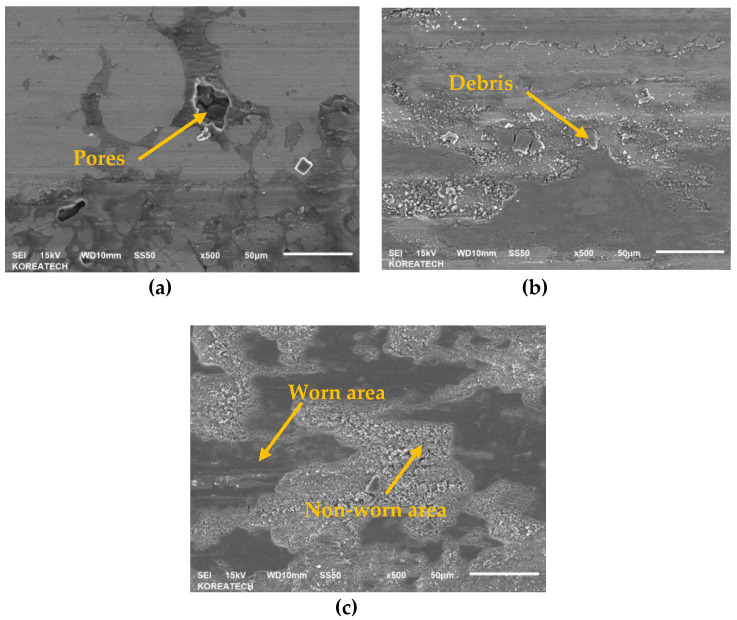
High-magnification SEM images of the wear track formed on the surface of the as-PM (**a**), UNSM-25 (**b**) and UNSM-300 (**c**) samples.

**Table 1 materials-13-03172-t001:** Chemical composition of the sintered Fe-based high C steel in (wt %).

C	Lub	MnS	Fe
0.70	0.80	0.45	Bal.

**Table 2 materials-13-03172-t002:** UNSM treatment parameters.

Frequency, kHz	Amplitude, μm	Speed, mm/min	Static Load, N	Interval, μm	Ball Diam., mm	Ball Material
20	30	2000	50	70	2.38	WC

**Table 3 materials-13-03172-t003:** Tribological test conditions.

Load, N	Frequency, Hz	Sliding Distance, m	Time, min	Temperature, ℃
10	1.33	25	26	23–25

**Table 4 materials-13-03172-t004:** FWHM (Full width at half maximum) values of the martensitic (200) martensitic peak.

As-PM	UNSM-25	UNSM-300
0.76	0.83	0.91

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
