# Peer review of "Better Surface Integrity and Tribological Properties of Steel Sintered by Powder Metallurgy"

_materials, 2020, doi:10.3390/ma13143172_

Round 1

Reviewer 1 Report

The study is focus on the ultra-sonic process using on steel sintered by P.M., and has great results in the surface integrity, friction and wear properties. Comments are listed as:

  1. Depth profiles of modification layers have to be shown in results. Because the depth of wear scare is lower than 10 micro-meters. I cannot check if the modification film is already broken. The other issue is the hardness measuring. How about the indentation depth under 300 gf indentation force? Substrate effect will let the hardness measuring value become lower than real value. Nano indentation can be considered in the study and the method can show elastic modulus at the same time. It can help in discussing the wear phenomena.
  2. After annealing to 300 degree C, precipitations around pores are coming out. What’s their compositions? They may affect your tribo-testing results.
  3. In the article section 3.3, authors mention that “According to the Hall-Patch 165 expression, the smaller the crystal size, the higher the hardness.” Please shown me the SEM results and discuss their grain sizes.

Author Response

We would like to thank the reviewers for the time and effort to review the manuscript and provide us with valuable comments and suggestions. We have revised the manuscript according to all the comments to the best of our ability and knowledge. Following are the details of the revisions:

Responses to comments

Reviewer #1:

The study is focus on the ultra-sonic process using on steel sintered by P.M., and has great results in the surface integrity, friction and wear properties. Comments are listed as:

Depth profiles of modification layers have to be shown in results. Because the depth of wear scare is lower than 10 micro-meters. I cannot check if the modification film is already broken. The other issue is the hardness measuring. How about the indentation depth under 300 gf indentation force? Substrate effect will let the hardness measuring value become lower than real value. Nano indentation can be considered in the study and the method can show elastic modulus at the same time. It can help in discussing the wear phenomena.

RESPONSE: In general, depth profiles of modification layers may reach up to about 200 µm depending on UNSM treatment parameters. As the depth of the wear track is lower than 10 micro-meters which is still remained within the depth of modification layer. The authors have mentioned about the depth profiles of modification layers in Section 3.5 citing a reference in the revised manuscript.

The indentation depth under 300 gf indentation force is about 35-38 micro-meters, which is also remained within the depth of modification layer. The authors have already tried a nano-indentation method for several years, but it is not applicable method due to the surface roughness of the samples, which is very rough to be measured by nano-indentation method. Thank you for your attention and suggestion.

After annealing to 300 degree C, precipitations around pores are coming out. What’s their compositions? They may affect your tribo-testing results.

RESPONSE: Perhaps, when the sample was treated by local heat treatment at 300 0C precipitations around pores will be come out. However, in this study, we did not focus on comprehensive characterization of the formation of precipitations around pores. Nevertheless, it is a good idea to characterize them in our next papers on the formation of precipitations around pores of MnS steel by UNSM at high temperature. Moreover, they may also effect on tribological properties of MnS steel. Thank you for your valuable comments.

In the article section 3.3, authors mention that “According to the Hall-Patch 165 expression, the smaller the crystal size, the higher the hardness.” Please shown me the SEM results and discuss their grain sizes.

RESPONSE: TEM images of the UNSM-25 and UNSM-300 samples showing the average grain size have been provided in the revised manuscript to support the Hall-Patch expression.

Reviewer 2 Report

Interesting experiments similar to shot peening process. There are some points to be clarified:

  • In the introduction (line 37), the authors say that PM has been recently applied for the fabrication of automotive components, but this happens since many decades ago.
  • Line. 45 Could the authors explain the relation between machinability and mechanical and tribological properties of the machined part? These PM products have high porosity, so MnS inclusions are not critical for the component performance
  • Do the samples undergo any thermal treatment after sintering?
  • How is S2PD affecting dimensional control of the component? The geometry of shock absorbers is much more complex than the ring geometry used in the experiments. Do you think this method is applicable to industrial conditions (millions of parts)?

  • Which is the WC grade used in the experiments? Is the WC ball wear taken into account?
  • Residual stresses produced by UNSM need to be measured and included in the paper, since this effect is dominant regarding crack formation and propagation.
  • Measurement of grain size after UNSM requires using Scherrer equation of similar (Rietveld method, etc). The authors cannot conclude that the loss of intensity in XRD peaks proves that a nanostructure is generated at the surface of the  specimens.
  • In Fig. 7, Peaks correspond to ferrite or martensite? No details are given about the cooling rate after sintering.

Author Response

RESPONSE TO REVIEWERS` COMMENTS

Materials

Ref.: Ms. No. materials-864167R1

Title: Better Surface Integrity and Tribological Properties of Steel Sintered by Powder Metallurgy

Authors: A. Amanov

We would like to thank the reviewers for the time and effort to review the manuscript and provide us with valuable comments and suggestions. We have revised the manuscript according to all the comments to the best of our ability and knowledge. Following are the details of the revisions:

Responses to comments

Reviewer #2:

Comments and Suggestions for Authors

Interesting experiments similar to shot peening process. There are some points to be clarified:

In the introduction (line 37), the authors say that PM has been recently applied for the fabrication of automotive components, but this happens since many decades ago.

RESPONSE: The correction has been made in the revised manuscript.

Line. 45 Could the authors explain the relation between machinability and mechanical and tribological properties of the machined part? These PM products have high porosity, so MnS inclusions are not critical for the component performance.

RESPONSE: The relationship between machinability, and mechanical and tribological properties The authors agree with the reviewer that the percentage of added MnS inclusions is too low, that`s why they are not critical for the component performance.

Do the samples undergo any thermal treatment after sintering?

RESPONSE: The samples do not undergo any thermal treatment after sintering. As mentioned in Section 2.2 that the samples have been treated by UNSM technology at 25 and 300 0C.

How is S2PD affecting dimensional control of the component? The geometry of shock absorbers is much more complex than the ring geometry used in the experiments. Do you think this method is applicable to industrial conditions (millions of parts)?

RESPONSE: The effect of UNSM technology on dimensional control of the components is very negligible based on the 3D measurement results (see below), that`s why the authors strongly believe that this UNSM method is applicable to industrial conditions. The authors need to mention here this technology is already being applied to various components in different industries. For example, bearings (inner and outer raceways), shafts, etc.

Which is the WC grade used in the experiments? Is the WC ball wear taken into account?

Residual stresses produced by UNSM need to be measured and included in the paper, since this effect is dominant regarding crack formation and propagation.

RESPONSE: WC dymet grade DF3 (ISO Code K01) has been used in the UNSM treatment. The wear WC ball has not been taken into account due to its high hardness of 2000 HV compared to the sintered steel. I would like to thank the reviewer because the wear of WC ball is an excellent question to take into account in our next researches.

Measurement of grain size after UNSM requires using Scherrer equation of similar (Rietveld method, etc). The authors cannot conclude that the loss of intensity in XRD peaks proves that a nanostructure is generated at the surface of the specimens.

RESPONSE: TEM images of the UNSM-25 and UNSM-300 samples showing the average grain size have been provided in the revised manuscript to conclude that the loss of intensity in XRD peaks also proves that a nanostructure is generated at the surface of the UNSM-25 and UNSM-300 samples.

In Fig. 7, Peaks correspond to ferrite or martensite? No details are given about the cooling rate after sintering.

RESPONSE: XRD patterns correspond to martensite derived from the as-PM, UNSM-25 and UNSM-300 samples. The cooling rate after sintering has been added to the revised manuscript.

Round 2

Reviewer 1 Report

The draft has been well modified, and no other additional comments.

Author Response

Thank you!